# OPEN VOCABULARY PANOPTIC SEGMENTATION WITH RETRIEVAL AUGMENTATION

## ABSTRACT

Given an input image and set of class names, panoptic segmentation aims to label each pixel in an image with class labels and instance labels. In comparison, Open Vocabulary Panoptic Segmentation aims to facilitate the segmentation of arbitrary classes according to user input. The challenge is that a panoptic segmentation system trained on a particular dataset typically does not generalize well to unseen classes beyond the training data. In this work, we propose a retrieval-augmented panoptic segmentation method that improves the performance of unseen classes. In particular, we construct a masked segment feature database using paired image-text data. At inference time, we use masked segment features from the input image as query keys to retrieve similar features and associated class labels from the database. Classification scores for the masked segment are assigned based on the similarity between query features and retrieved features. The retrieval-based classification scores are combined with CLIP-based scores to produce the final output. We incorporate our solution with a previous SOTA method (FC-CLIP). When trained on COCO, the proposed method demonstrates 30.9 PQ, 19.3 mAP, 44.0 mIoU on the ADE20k dataset, achieving +4.5 PQ, +2.5 mAP, +10.0 mIoU absolute improvement over the baseline.

## 1 INTRODUCTION

Panoptic segmentation (Kirillov et al., 2019) is a computer vision task that combines semantic segmentation and instance segmentation. Semantic segmentation (Long et al., 2015) labels every pixel in an image with a class category, such as "tree" or "car." Instance segmentation (Bolya et al., 2019) differentiates between individual objects of the same class (1st car, 2nd car). Panoptic segmentation unifies these tasks to label every pixel with a class label and identify distinct objects within the same category with an instance label. This method is valuable in fields like autonomous driving (Feng et al., 2020) and robotics (Milioto & Stachniss, 2019), where detailed scene understanding is crucial. A key challenge for traditional panoptic segmentation is the need for highly granular pixel-level data annotation. Lack of data limits the number of possible classes for panoptic segmentation, making the system closed-vocabulary (Ding et al., 2023).

Open vocabulary panoptic segmentation (Ding et al., 2023; Xu et al., 2023c; Yu et al., 2024) is an advanced version of the traditional panoptic segmentation task that extends its capabilities to identify and label objects from a potentially unlimited set of classes. Unlike standard panoptic segmentation which relies on a fixed set of known classes, open vocabulary segmentation allows the system to recognize and categorize objects even if they haven't been specifically included in the training dataset.

Recent methods for open vocabulary segmentation (Ding et al., 2023; Xu et al., 2022b; Liang et al., 2023; Xu et al., 2023c; Yu et al., 2024) involves a two-stage framework. The first step is to generate a class-agnostic mask proposal and the second step is to leverage pre-trained vision language models (e.g., CLIP (Radford et al., 2021)) to classify masked regions. In this approach, the input class descriptions are encoded with a CLIP text encoder and the masked image region is encoded with a CLIP vision encoder. The masked region is classified based on the cosine similarity of masked image features and class-related text features. CLIP has shown the ability to improve open vocabulary performance because it is pre-trained to learn joint image-text feature representation from large-scale internet data. However, the performance of the CLIP vision encoder suffers from a limitation

when we encode a masked image instead of a natural image. This poor quality of encoded features hurts open vocabulary segmentation performance (Liang et al., 2023).

In this work, we address the bottleneck mentioned above in the context of open vocabulary panoptic segmentation. In order to mitigate the domain shift between the natural image feature and the masked image feature, we propose a retrieval-augmented approach for panoptic segmentation. Specifically, we first use large-scale image-text pairs to construct a feature database with associated text labels for the masked regions. Then during inference time, the masked region feature extracted from the input image is used as a retrieval key to retrieve similar features and associated class labels from the database. The masked region is classified based on the similarity between the retrieval key and retrieval targets. Since both the retrieval key and retrieval target use a CLIP vision encoder on masked regions, the proposed approach does not suffer from the domain shift between the natural image feature and the masked image feature. We combine this retrieval-based classification module with the CLIP-based classification module to improve open vocabulary panoptic segmentation performance. Our contributions are as follows:

- We proposed a retrieval-augmented panoptic segmentation approach that tackles the domain shift between the natural image feature and masked image feature with respect to the CLIP vision encoder. The proposed approach can incorporate new classes in the panoptic segmentation system simply by updating the feature database in a fully training-free manner. Moreover, the feature database can be constructed from paired image-text data which is widely available for thousands of classes.

- We demonstrate that the proposed system can improve open vocabulary panoptic segmentation performance in both training-free setup (+5.2 PQ) and cross-dataset fine-tuning setup (+ 4.5 PQ, COCO→ADE20k).

## 2 RELATED WORK

**Fully Supervised** Fully supervised methods typically involve training or fine-tuning the system on a dataset with pixel-level annotations (Li et al., 2022; Ghiasi et al., 2022; Xu et al., 2022c; Luo et al., 2023a). Ding et al. (2023) use a trainable relative mask attention module to produce robust masked segment features from a frozen CLIP backbone. Xu et al. (2023a) proposes combining the internal representation of pretrained text-to-image diffusion models and discriminative image-text models for open vocabulary panoptic segmentation. Liang et al. (2023) fine-tune a CLIP backbone to improve alignment between text representation and masked image representation. Xu et al. (2023c) use a student-teacher self-training to improve mask generation for unseen classes and fine-tune CLIP to improve query feature representation. Yu et al. (2024) use a frozen CNN-based CLIP backbone for both mask proposal generation as well as classification.

**Weakly Supervised** Weakly supervised methods are trained on image-level annotations (Xu et al., 2022a; Liu et al., 2022; Zhou et al., 2022; Xu et al., 2023b). Luo et al. (2023b) train the system on image-text pairs using a semantic group module to aggregate patches with learnable image regions. He et al. (2023) use self-supervised pixel representation learning guided by CLIP image-text alignment for semantic segmentation. Mukhoti et al. (2023) propose patch-level contrastive learning that learns alignment between visual patch tokens and text tokens. This approach generalizes to the open vocabulary setting without any training on pixel-level annotations. Wang et al. (2024b) combine the spatial understanding of Segment Anything Model (SAM) (Kirillov et al., 2023) and semantic understanding of CLIP for open vocabulary semantic segmentation. They use continual learning and knowledge distillation methods to ensure the resulting model retains the capabilities of the original models.

**Training Free** Training-free methods typically exploit pretrained models (e.g. CLIP) for open vocabulary segmentation without any fine-tuning on pixel-level or image-level annotations (Wang et al., 2024c; Tang et al., 2024; Wang et al., 2024a). Shin et al. (2022) construct a database of reference image segments using CLIP. During inference, the reference images are used for segmenting relevant segments from the input image. Karazija et al. (2024) generate synthetic reference images using a text-to-image diffusion model and perform segmentation by comparing input images with synthetic references. Wysoczańska et al. (2024) encodes small image patches separately to the vi-

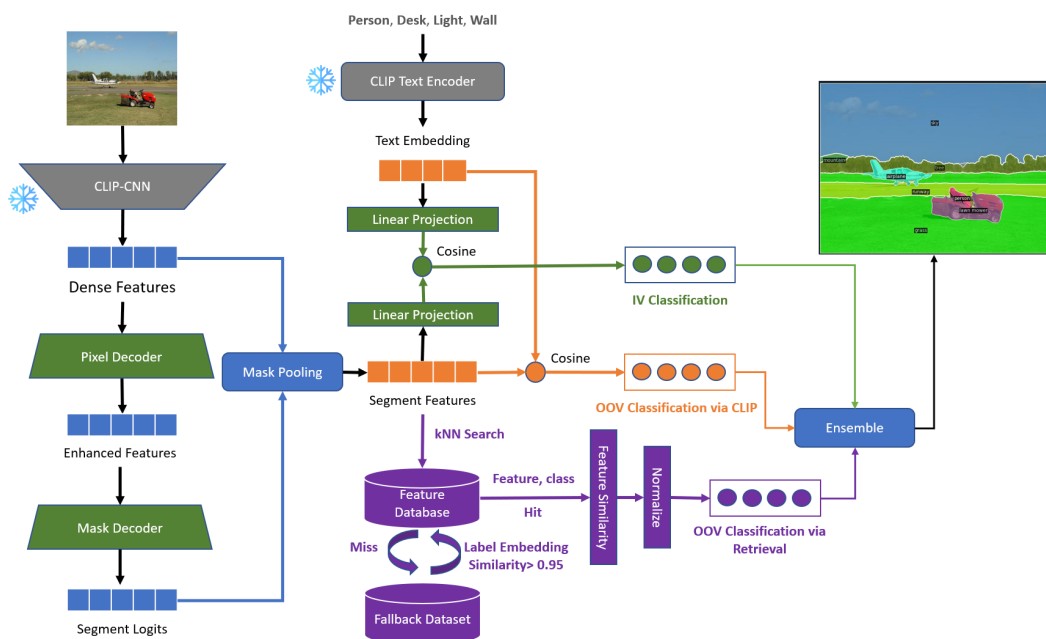

Figure 1: Overview of the open vocabulary panoptic segmentation method (cross-dataset)

sion encoder and computes class-specific similarity for an arbitrary number of classes. Then they perform patch aggregation, up-sampling, and foreground-background segmentation to produce segmentation for unseen classes. Gui et al. (2024) construct a feature database of masked segment features and use retrieval to perform panoptic segmentation on unseen categories. There are two key differences between their approach and our proposed method. Firstly, Gui et al. (2024) uses one visual encoder for mask proposal generation and masked segment classification and a separate visual encoder to construct retrieval key features. We demonstrate that a single CLIP backbone with mask pooling can be used for all three tasks: mask proposal generation, retrieval key generation, and masked segment classification. Secondly, Gui et al. (2024) rely on ground truth masks for constructing the feature database so their proposed approach cannot be extended to a new dataset where pixel-level annotation is unavailable. We use open vocabulary object detection combined with SAM for constructing the feature database and demonstrate that our approach achieves performance improvement by exploiting a completely different dataset with only image-level annotations.

## 3 METHODOLOGY

### 3.1 CROSS DATASET PANOPTIC SEGMENTATION

In the cross-dataset variant of open vocabulary panoptic segmentation, the system is fine-tuned on one dataset (e.g. COCO) and evaluated on another dataset (ADE20k) with some unseen classes. Our cross-dataset method is based on FC-CLIP (Yu et al., 2024) where a mask proposal generator and mask decoder are fine-tuned on COCO (Lin et al., 2015). The overview of the system is shown in Figure 1.

**Shared Backbone**    Similar to FC-CLIP, we use a frozen CNN-based CLIP backbone. The backbone is shared between the mask generation and segment classification. Yu et al. (2024) have demonstrated that CNN-based CLIP backbone is a more robust variation in image resolution. We use the ConvNeXt-Large variant of CLIP backbones from OpenCLIP (Cherti et al., 2023). The model is trained on the LAION-2B dataset (Schuhmann et al., 2022). The CLIP backbone converts the input image to patch-specific dense features which is used for mask generation and segment classification.

**Mask Proposal Generation**    The mask proposal generator is based on Mask2former (Cheng et al., 2022). A pixel decoder is used for enhancing dense features from the CLIP backbone. The en-

hanced features and class-related queries are fed to a series of mask decoders. The mask decoders are equipped with self-attention, masked cross-attention, and a feed-forward network. Finally, the segmentation logits are produced via matrix multiplication between class queries and transformed pixel features.

**In Vocabulary Classification**  The in-vocabulary classification path is shown in green in Figure 1. The dense features are computed from the input image feature and mask proposals using mask pooling. Dense features for masked regions and class name embeddings are projected to the same embedding space using linear projection. The linear projection parameters for in-vocabulary classifiers are fine-tuned on COCO. The classification scores are obtained based on cosine similarity between class embeddings and masked segment features.

**Out-of-vocabulary Classification Via Retrieval**  The retrieval-based classification path is shown in violet in Figure 1. The retrieval module uses masked segment features as retrieval keys to perform approximate nearest neighbor search in the feature database. The output is a set of distance scores between the retrieval key and retrieval targets and associated class labels. The distance scores are normalized using min-max normalization and subtracted from one. This step produces retrieval-based classification scores. In case any of the user-provided class names are missing in the feature database, we retrieve image samples for those input classes from a secondary image dataset. The label matching between datasets is performed with CLIP text embedding of class names with similarity score $> 0.95$.

**Out-of-vocabulary Classification Via CLIP**  Similar to FC-CLIP, we have a CLIP-only segment classifier. This is helpful in case the feature database does not have similar features compared to the segment features. The classification is performed using cosine similarity between segment features and class name embeddings. Unlike in-vocabulary classifiers, the features do not go through fine-tuned linear projection layers.

**Ensemble**  Let's assume $C$ is the set of classes for prediction and $C_{train}$ is the set of classes in the fine-tuning dataset. Let $s^i_{clip}, s^i_{ret}, s^i_{iv}$ be classification scores for class $i$ using CLIP, retrieval and in-vocabulary classifier. The scores from the three classification pipelines are combined as follows, where $\alpha, \beta, \gamma$ are hyper-parameters.

$$s^i_{oov} = s^i_{ret} \times \gamma + s^i_{clip} \times (1 - \gamma)$$
$$s^i = s^i_{oov} \times \alpha + s^i_{iv} \times (1 - \alpha) \quad \texttt{if } i \in C_{train}$$
$$s^i = s^i_{oov} \times \beta + s^i_{iv} \times (1 - \beta) \quad \texttt{if } i \notin C_{train}$$

## 3.2 Training Free Panoptic Segmentation

In training free variant of open vocabulary panoptic segmentation, none of the system components are fine-tuned on pixel-level panoptic annotations. We use an open vocabulary objection detection model and SAM for mask proposal generation. The segment classification was performed with CLIP and retrieval. The overview of the system is shown in Figure 2.

**Mask Proposal Generation**  Given an input image and a list of classes, we use Grounding DINO (Liu et al., 2024) to detect bounding boxes associated with each class. All bounding boxes detected with a minimum confidence threshold are retained. The bounding boxes are passed to SAM for generating class-aware masks. The outputs of SAM are used as class-agnostic mask proposals. All potential classes for panoptic segmentation are passed to the object detection method and confidence-based filtering is performed to prune absent classes.

**Dense Feature for Masked Regions**  A CLIP backbone is used to extract dense features from the input image. The mask proposals from the previous step are used to extract masked image regions from the image-level dense features. We use mask pooling to convert image-level dense features to region-level dense features.

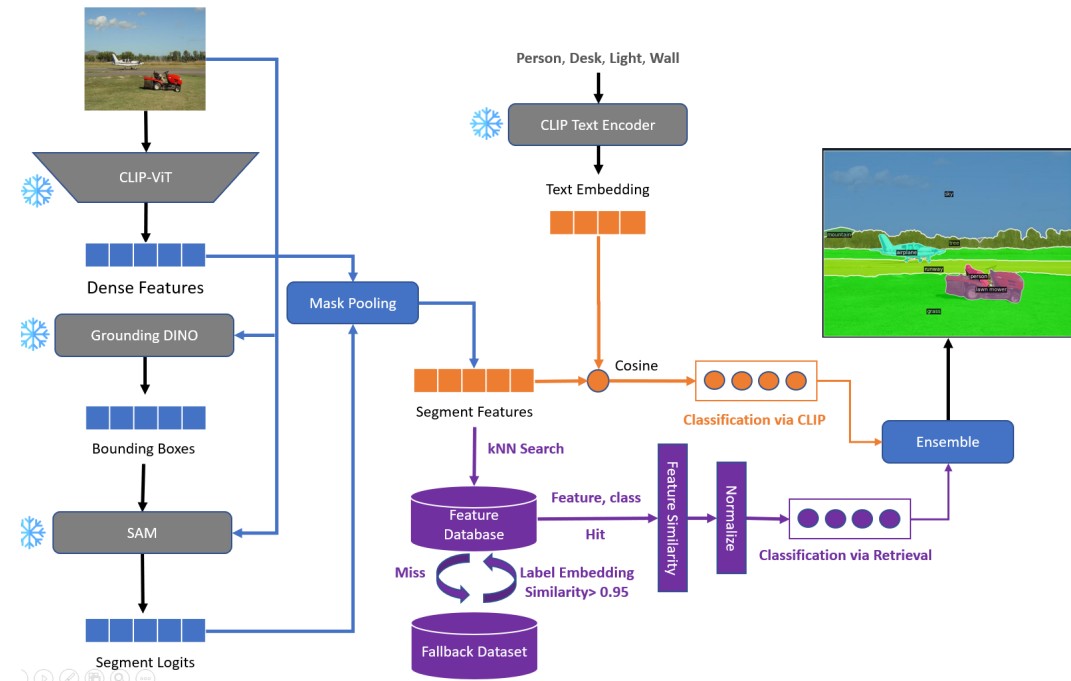

Figure 2: Overview of the open vocabulary panoptic segmentation method (training free)

**Classification with CLIP** The input class names are encoder with CLIP text encoders. The cosine similarity between CLIP text embeddings and dense features for each mask region is used to classify each masked region.

**Retrieval-based Classification** For each dense feature associated with a masked region, we perform an approximate nearest neighbor search in the feature database to retrieve the most similar features and associated class labels. The retrieval distances are normalized with min-max normalization and subtracted from one to produce classification scores.

**Ensemble** Let's assume $C$ is the set of classes for prediction. Let $s^i_{clip}, s^i_{ret}$ be classification scores for class $i$ using CLIP and retrieval. The scores from the two classification pipelines are combined as follows, where $\gamma$ is a hyper-parameter.

$$s^i = s^i_{ret} \times \gamma + s^i_{clip} \times (1 - \gamma)$$

### 3.3 FEATURE DATABASE CONSTRUCTION

The objective of the database construction step is to take a paired image-text dataset as input and convert it into a database of masked segment features and associated class labels. The database construction has four steps, namely object detection, mask generation, dense feature generation, and mask pooling. The overview of the process is shown in Figure 3.

**Object Detection** In this step, an image and class labels present in the image are fed to an open vocabulary object detection method. The output is a bounding box associated with each class present in the image. We use the SOTA open vocabulary object detection method Grounding DINO (Liu et al., 2024).

**Mask Generation** In this step, the input image and associated bounding box prompts are fed to SAM (Kirillov et al., 2023) for mask generation. Even though SAM can generate masks without class-aware bounding boxes, the resulting masks often break up a single class (e.g. car) into multiple masks (e.g. wheel, car body, window). An example of this phenomenon is shown in Figure 4. The

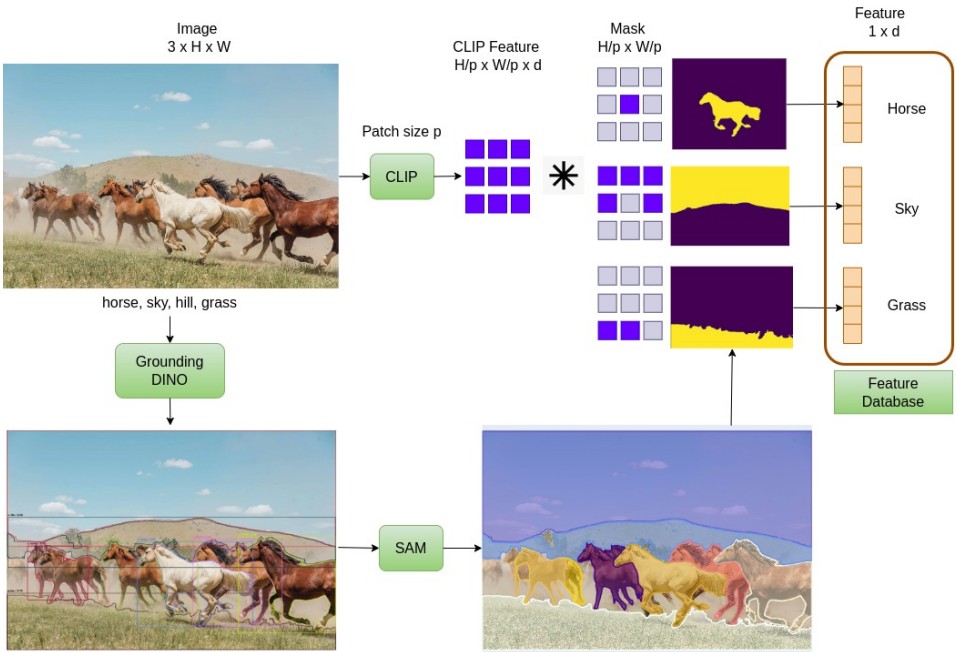

Figure 3: Overview of feature database construction

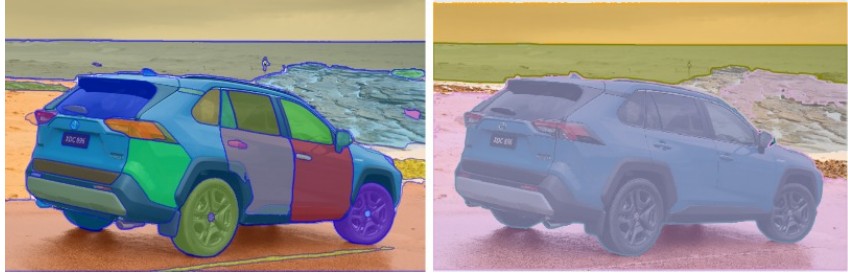

Figure 4: a) Left: mask generation with SAM point prompt sampling b) Right: class aware mask generation with Grounding DINO + SAM

class-aware masks generated in the previous step ensure that the SAM can generate high-quality masks for each class present in the image.

**Dense Feature Generation**    We use CLIP to extract dense features from an image. Let's assume that the input image has shape $3 \times H \times W$, the patch size of CLIP is $p$, and the dimension of the dense feature is $d$. The shape of the output dense feature is $\frac{H}{p} \times \frac{W}{p} \times d$.

**Mask Pooling**    Mask pooling operation involves taking dense features associated with the whole image and generating mask-specific dense features based on generated masks in the second step. This way we don't have to encode each masked segment using CLIP separately which can be computationally expensive (Yu et al., 2024). The mask pooling operation generates a $d$ dimensional feature vector for each masked segment. These features and associated class labels are added to the database.

## 4    EVALUATION

**Setup**    The training-free setup does not use any panoptic segmentation annotations. The cross-dataset setup is fine-tuned on COCO panoptic annotations. For constructing the retrieval fea-

Table 1: Open vocabulary panoptic segmentation performance in training free setup

| Mask Proposal | Region Classification | Image Encoder | Database | PQ | mAP | mIoU |
|---|---|---|---|---|---|---|
| Grounding DINO + SAM | CLIP Baseline | CLIP-ViT-large | ADE20k | 0.109 | 0.069 | 0.138 |
| Grounding DINO + SAM | Retrieval Baseline | CLIP-ViT-large | ADE20k | 0.158 | 0.098 | 0.215 |
| Grounding DINO + SAM | Retrieval + CLIP | CLIP-ViT-large | ADE20k | **0.161** | **0.103** | **0.222** |

Table 2: Open vocabulary panoptic segmentation performance in cross-dataset setup

| Method | Image Encoder | Database | Fine-tuning | PQ | mAP | mIoU |
|---|---|---|---|---|---|---|
| FC-CLIP | CLIP-ConvNeXt-large | ADE20k | COCO | 0.264 | 0.168 | 0.340 |
| FC-CLIP + retrieval | CLIP-ConvNeXt-large | ADE20k | COCO | **0.309** | **0.193** | **0.440** |
| FC-CLIP + retrieval | CLIP-ConvNeXt-large | Google Open Image | COCO | 0.283 | 0.177 | 0.383 |

ture database, we use the ADE20k (Zhou et al., 2019) train set and Google Open Image dataset (Kuznetsova et al., 2020) in separate settings. The evaluations are reported on the ADE20k validation set. Out of 150 classes in the ADE20k validation set, 70 are present in COCO. These classes serve as in-vocabulary classes and the rest of the classes are out-of-vocabulary. We experiment with different CLIP backbones such as CLIP-ViT-base, CLIP-ViT-large, CLIP-ConvNeXt-large. We use Grounding-DINO-base for object detection and SAM-ViT-base for segmentation. We experiment with three different mask proposal methods such as ground truth mask, point prompt grid sampling with SAM, and Grounding DINO with SAM.

**Baseline and Metrics**  We use CLIP baseline for the training-free setup and FC-CLIP baseline in the cross-dataset setup. For hyper-parameters in the FC-CLIP baseline, we the the same configuration used by Yu et al. (2024), setting $\alpha = 0.4, \beta = 0.8$. We use panoptic quality (PQ), mean intersection over union (mIoU), and mean average precision (mAP) as evaluation metrics.

**Results**  Retrieval-augmented classification improves performance in both training-free setup and cross-dataset fine-tuning setup. In the training-free setup, the proposed method (retrieval + CLIP) achieves 47% relative improvement in PQ (+5.2 absolute) and 60% relative improvement (+8.4 absolute) in mIoU (shown in Table 1). In the cross-dataset setup, the proposed method achieves 17% relative improvement in PQ (+4.5 absolute) and 29% relative improvement (+10.0 absolute) in mIoU. The proposed method also improves performance when the retrieval features are constructed from a completely different dataset such as Google Open image, as shown in Table 2.

We demonstrate the impact of the mask proposal generator in Table 3. The system achieves a PQ of 27.2 with a ground truth mask with a CLIP-ViT-large backbone. Automatic mask generation with SAM performs poorly with a PQ of 7.8. The reason is that SAM is trained for interactive input with humans in the loop. Without human input, SAM masks are not class-aware. SAM may break up a single object into multiple fine masks as shown in Figure 4. We mitigate this issue by using open vocabulary object detection to construct class-aware bounding boxes and feeding them to SAM. This approach improves PQ to 16.1 in the training-free setup. The hyper-parameter tuning for ensemble coefficients is shown in Table 4. We find best performance with $\alpha = 0.4, \beta = 0.7, \gamma = 0.3$.

## 5  CONCLUSIONS

In this work, we exploit a retrieval-based method for improving open vocabulary panoptic segmentation. We construct a visual feature database using paired image-text data. During inference, we use masked segment features from the input image as query keys to retrieve similar features and associated class labels from the database. Classification scores for the masked segment are assigned based on the similarity between query features and retrieved features. The retrieval-based classification scores are combined with CLIP-based scores to produce the final prediction. The proposed approach improves PQ from 26.4 to 30.9 on ADE20k when fine-tuned on COCO. Even though the proposed method achieves reasonable performance in an open vocabulary setting, it remains vulnerable to the quality of mask proposal generation. Future work may focus on improving the quality of mask proposal generation for unknown classes.

Table 3: Impact of mask proposal quality. The results are shown for the training-free setup.

| Mask Proposal | Region Classification | Image Encoder | Database | PQ | mAP | mIoU |
|---|---|---|---|---|---|---|
| Ground Truth | CLIP Baseline | CLIP-ViT-base | ADE20k | 0.160 | 0.092 | 0.224 |
| Ground Truth | Retrieval Baseline | CLIP-ViT-base | ADE20k | 0.210 | 0.130 | 0.254 |
| Ground Truth | Retrieval + CLIP | CLIP-ViT-base | ADE20k | **0.211** | **0.133** | **0.276** |
| Grid Sampling + SAM | CLIP Baseline | CLIP-ViT-base | ADE20k | 0.042 | 0.025 | 0.059 |
| Grid Sampling + SAM | Retrieval Baseline | CLIP-ViT-base | ADE20k | 0.048 | 0.032 | 0.065 |
| Grid Sampling + SAM | Retrieval + CLIP | CLIP-ViT-base | ADE20k | **0.052** | **0.034** | **0.069** |
| Grounding DINO + SAM | CLIP Baseline | CLIP-ViT-base | ADE20k | 0.090 | 0.055 | 0.123 |
| Grounding DINO + SAM | Retrieval Baseline | CLIP-ViT-base | ADE20k | 0.117 | 0.071 | 0.150 |
| Grounding DINO + SAM | Retrieval + CLIP | CLIP-ViT-base | ADE20k | **0.127** | **0.075** | **0.173** |
| Ground Truth | CLIP Baseline | CLIP-ViT-large | ADE20k | 0.217 | 0.139 | 0.291 |
| Ground Truth | Retrieval Baseline | CLIP-ViT-large | ADE20k | 0.272 | 0.165 | 0.346 |
| Ground Truth | Retrieval + CLIP | CLIP-ViT-large | ADE20k | **0.284** | **0.173** | **0.394** |
| Grid Sampling + SAM | CLIP Baseline | CLIP-ViT-large | ADE20k | 0.056 | 0.035 | 0.074 |
| Grid Sampling + SAM | Retrieval Baseline | CLIP-ViT-large | ADE20k | 0.066 | 0.039 | 0.086 |
| Grid Sampling + SAM | Retrieval + CLIP | CLIP-ViT-large | ADE20k | **0.078** | **0.042** | **0.112** |
| Grounding DINO + SAM | CLIP Baseline | CLIP-ViT-large | ADE20k | 0.109 | 0.069 | 0.138 |
| Grounding DINO + SAM | Retrieval Baseline | CLIP-ViT-large | ADE20k | 0.158 | 0.098 | 0.215 |
| Grounding DINO + SAM | Retrieval + CLIP | CLIP-ViT-large | ADE20k | **0.161** | **0.103** | **0.222** |

Table 4: Hyper-parameter tuning, cross dataset setup

| $\alpha$ | $\beta$ | $\gamma$ | PQ | $\alpha$ | $\beta$ | $\gamma$ | PQ |
|---|---|---|---|---|---|---|---|
| 1.0 | 1.0 | 0.3 | 0.248 | 0.4 | 0.7 | 0.5 | 0.278 |
| 0.5 | 0.7 | 0.3 | 0.303 | 0.4 | 0.7 | 0.4 | 0.297 |
| 0.4 | 0.9 | 0.3 | 0.299 | 0.4 | 0.7 | 0.3 | **0.309** |
| 0.4 | 0.8 | 0.3 | 0.303 | 0.4 | 0.7 | 0.2 | 0.309 |
| 0.4 | 0.7 | 1.0 | 0.254 | 0.4 | 0.7 | 0.1 | 0.299 |
| 0.4 | 0.7 | 0.7 | 0.278 | 0.4 | 0.7 | 0.0 | 0.264 |
| 0.4 | 0.7 | 0.6 | 0.288 | 0.3 | 0.7 | 0.3 | 0.305 |

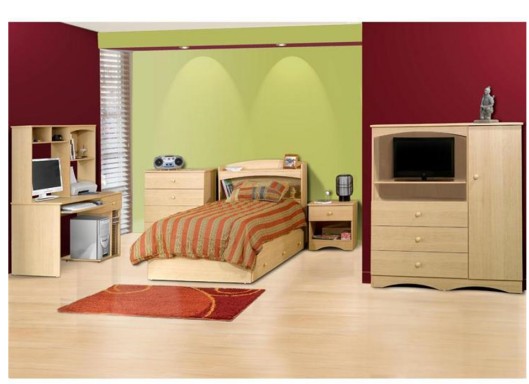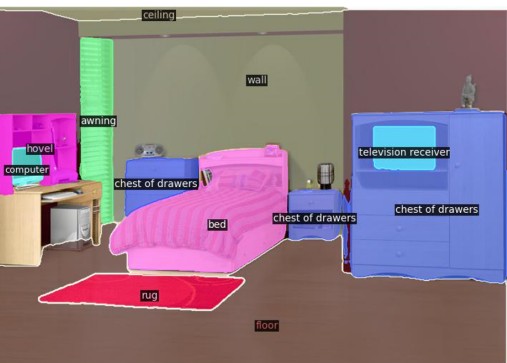

Figure 5: Case Study 1. Out-of-vocabulary class: computer, chest of drawers.

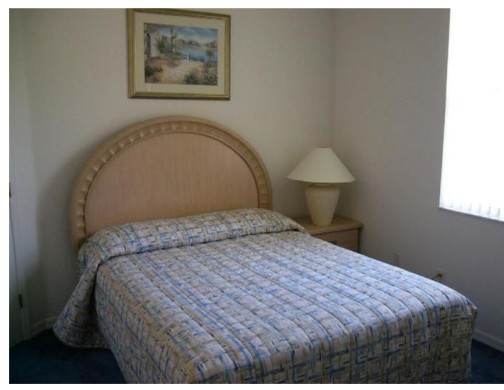 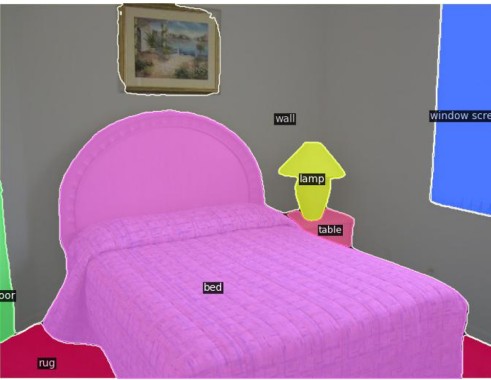

Figure 6: Case Study 2. Out-of-vocabulary class: lamp, window screen

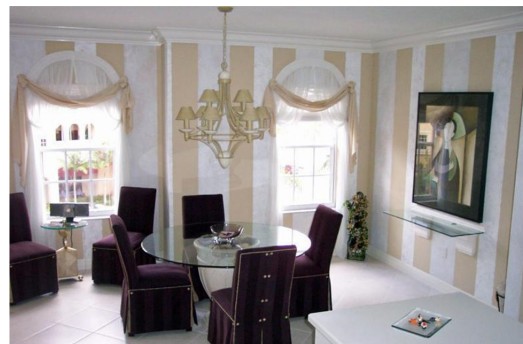 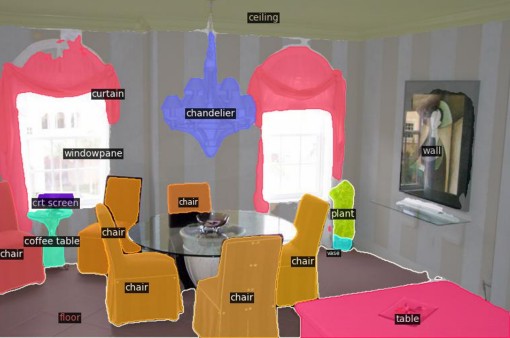

Figure 7: Case Study 3. Out-of-vocabulary class: chandelier, coffee table.

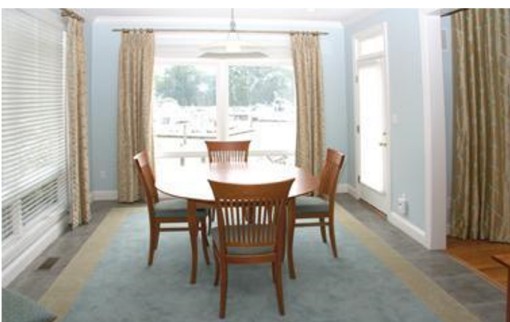 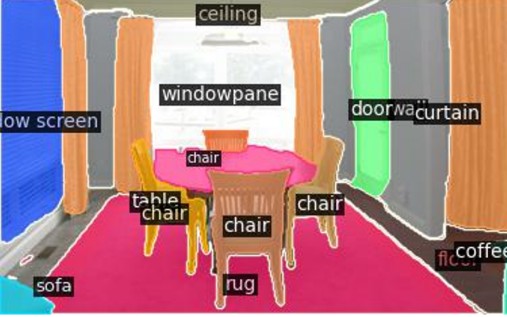

Figure 8: Case Study 4. Out-of-vocabulary class: window screen

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
