# OpenReview forum: "Open Vocabulary Panoptic Segmentation With Retrieval Augmentation"
_ICLR.cc/2025/Conference — Submitted to ICLR 2025_

### Official Review · Reviewer_GoTi · 2024-10-21

**Soundness:** 2
**Presentation:** 3
**Contribution:** 2
**Rating:** 3
**Confidence:** 4

**Summary:**

This paper introduces a retrieval-based method to enhance the performance of open vocabulary panoptic segmentation by constructing a feature database from paired image-text data. During inference, the model uses masked segment features from the input image to query the database for similar features and associated class labels, which are then combined with CLIP-based scores. This approach leads to improvements in Panoptic Quality in both training-free and cross-dataset settings.

**Strengths:**

The paper explains related concepts clearly and details the methodology comprehensively, making the overall article easy to understand.

**Weaknesses:**

1. The novelty of the paper is limited, primarily building upon the feature retrieval idea from Gui et al.[1]. Compared to Gui et al. [1]., the main modifications only include using a single CLIP backbone instead of two backbone. Please explain how these contributions can meet the strict requirements of top-level conferences.

2. The authors use open vocabulary object detection combined with SAM to build the feature database, which limits the model's performance to the capabilities of the object detection component. Please explain how to handle classes that are not included in both the feature database and the fallback dataset during inference, or discuss the limitations of their approach for truly open-vocabulary scenarios.

3. The definitions of IV Classification and OOV Classification are confusing. Why is it considered that the segment features and text embedding after linear projection in Figure 1 are equivalent to IV Classification? Please provide a more detailed explanation of the distinction between these two classifications and why the linear projection is significant for IV Classification.

4. The experimental section lacks a critical component: comparisons with state-of-the-art methods, such as Gui et al. [1]., HIPIE [2], ODISE [3], OPSNet [4]. Please explain why these specific comparisons are not included and how your method compares theoretically to these state-of-the-art approaches.

5. How does this method perform on open vocabulary semantic segmentation tasks, such as testing on ADE20K-847, ADE20K-150, Pascal Context-459.

6. The paper claims to achieve performance improvement by utilizing a completely different dataset with only image level annotations. However, using the ADE20K training set to construct a feature database and evaluating it on the ADE20K validation set in the experiment lacks persuasiveness for open vocabulary. Please clarify how to ensure the open vocabulary nature when using the same dataset for both feature database construction and evaluation.

7. There is irrelevant content in the lower-left corner of Figure 2. Please redraw the figure and ensure that the image is complete and free from irrelevant content


reference:

[1] Zhongrui Gui, Shuyang Sun, Runjia Li, Jianhao Yuan, Zhaochong An, Karsten Roth, Ameya Prabhu, and Philip Torr. knn-clip: Retrieval enables training-free segmentation on continually expanding large vocabularies, 2024. URL https://arxiv.org/abs/2404.09447.

[2] Wang X, Li S, Kallidromitis K, et al. Hierarchical open-vocabulary universal image segmentation[J]. Advances in Neural Information Processing Systems, 2024, 36.

[3] Xu J, Liu S, Vahdat A, et al. Open-vocabulary panoptic segmentation with text-to-image diffusion models[C]//Proceedings of the IEEE/CVF Conference on Computer Vision and Pattern Recognition. 2023: 2955-2966.

[4] Chen X, Li S, Lim S N, et al. Open-vocabulary panoptic segmentation with embedding modulation[C]//Proceedings of the IEEE/CVF International Conference on Computer Vision. 2023: 1141-1150.

**Questions:**

1. What is the difference between IV Classification and OOV Classification cia CLIP in cross-dataset panoptic segmentation? What is the significance of this distinction? From Figure 1, it appears that the former only differs from the latter by including a linear projection.
2. What is the fallback dataset, and how the author build it?

**Details Of Ethics Concerns:**

There is no ethics concerns.

---

> ### Author Response · Authors · 2024-11-23
> **Rebuttal to reviewer GoTi**
>
> **Difference between IV Classification and OOV Classification via CLIP**
>  The IV classification system includes components that are fine-tuned on a panoptic segmentation dataset with pixel-level annotations. The trainable components are shown in green in Figure 1. This system is specifically fine-tuned on COCO. If the input class names are within the COCO vocabulary, the IV system demonstrates superior performance due to fine-tuning. This helps the system handle the most frequent classes (person, cars, sky, etc.) and even generalizes to a new dataset (e.g., ADE20k). OOV classification via CLIP is applied when the input class names are not part of the fine-tuning dataset. The OOV system does not have trainable linear projection layers to improve text and image embedding beyond the vanilla CLIP.
>
> **Fallback Dataset**
>  The fallback dataset can be any paired image-text dataset that includes sample images for novel target classes. An ideal candidate is the Google Open Image dataset v7, which has 61 million image-level labels across more than 20,000 classes. We use the fallback dataset to demonstrate that our feature database can be easily extended to novel classes by utilizing large-scale image-text data, which is widely available for a large number of classes.
>
> If the input class names include a class name that is not represented in the feature database, we can retrieve image samples for that novel class from the fallback dataset, compute features, and add them to the database. Thus, our proposed method can be easily extended to novel classes with minimal effort.

---

### Official Review · Reviewer_xAjY · 2024-10-24

**Soundness:** 1
**Presentation:** 2
**Contribution:** 2
**Rating:** 3
**Confidence:** 4

**Summary:**

This paper enhances open-vocabulary panoptic segmentation by leveraging retrieval augmentation to address the challenges of classifying unseen objects. The authors propose a framework that integrates masked segment features with a retrieval-based method to improve performance for unseen classes. The model builds a feature database using paired image-text data and retrieves similar features during inference to classify masked segments. These retrieval-based scores are combined with CLIP-based scores to enhance accuracy. When applied to FC-CLIP, the proposed method demonstrates improvements in unseen classes on the ADE20k dataset.

**Strengths:**

Applying Retrieval Augmentation to vision tasks is a promising direction. The proposed way of constructing a database is interesting.

**Weaknesses:**

1. While the method builds on FC-CLIP, the authors do not provide an introduction to FC-CLIP, which makes the paper hard to follow during reading.
2. The feature database should be introduced prior to discussing the retrieval method to improve the flow and clarity of the paper.
3. Since retrieval augmentation is intended to be a more general approach, the paper would benefit from presenting a more generalized framework to reflect its broader applicability.
4. The method of constructing the feature database itself serves as a strong baseline. How does the performance of the proposed retrieval-augmentation approach compare to Grounding DINO?
5. The paper lacks essential evaluations (the method is only evaluated on a single dataset with a single base model) and ablation studies.

**Questions:**

Please refer to the weaknesses. I think the current version is not ready for publication. More experiment results are expected.

---

### Official Review · Reviewer_X1Um · 2024-11-02

**Soundness:** 2
**Presentation:** 2
**Contribution:** 2
**Rating:** 5
**Confidence:** 5

**Summary:**

The paper presents a novel approach to address the challenge of segmenting arbitrary classes in images, a task known as open vocabulary panoptic segmentation. The authors propose a retrieval-augmented method that leverages a masked segment feature database constructed from image-text pairs. During inference, the system uses masked segment features from the input image to retrieve similar features and class labels from the database, combining these retrieval-based classification scores with CLIP-based scores to produce the final output. The method is evaluated on the ADE20k dataset and shows significant improvements over the baseline, particularly when fine-tuned on the COCO dataset, with absolute improvements of +4.5 PQ, +2.5 mAP, and +10.0 mIoU.

**Strengths:**

1. The paper introduces a creative solution to the open vocabulary panoptic segmentation problem by combining retrieval-based classification with CLIP, which is an original approach not commonly seen in the literature.
2. The paper is well-structured, with clear explanations of the methodology.

**Weaknesses:**

1. While the paper demonstrates improvements over the baseline, it does not provide a direct comparison with other state-of-the-art methods in the field, which could provide additional context for the significance of the results.
2. The discussion on how the proposed method generalizes to unseen classes could be expanded, as this is a critical aspect of open vocabulary segmentation.
3. The paper could further discuss the limitations of the retrieval-augmented approach, especially regarding the reliance on the quality of the feature database and the potential scalability issues as the number of classes increases.

**Questions:**

1. Could the authors elaborate on how their method generalizes to completely unseen classes that are not represented in the feature database?
2. As the number of classes in the feature database grows, how does the retrieval process scale in terms of computational resources and accuracy?
3. Are there any plans to compare the proposed method with other leading approaches in the field to contextualize the improvements?
4. The paper mentions that the quality of mask proposal generation is crucial. Could the authors provide more details on how variations in mask quality affect the final segmentation results?
5. Is there potential to integrate this method with other modalities, such as depth information or video data, to further improve performance?

---

> ### Author Response · Authors · 2024-11-23
> **Rebuttal to reviewer X1Um**
>
> **Generalizing to Unseen Classes**
>  If a class is not represented in the feature database, then the performance of the proposed system on that class is expected to be similar to a CLIP-only baseline. However, extending the feature database is very easy since we only require paired image-text data with class-level annotations. This is widely available for a large number of classes. The YFCC100M dataset alone contains 100 million paired image-text samples. Similarly, the Google Open Image dataset v7 has 61 million image-level labels on more than 20,000 classes. Our proposed method can easily generalize to unseen classes by collecting sample images for a new target class from large-scale paired image-text data.
>
> **Scaling**
>  Since we implement approximate nearest neighbor search with FAISS, the proposed method can scale to millions of images with minimal additional overhead compared to the baseline. We refer the reviewer to the time complexity analysis in the FAISS [1] paper for more details.
>
> **Comparison with Other Leading Approaches**
>  We plan to compare our method with Possam [2], which outperforms FC-CLIP by improving mask proposal generation with SAM using trainable components.
>
> **Impact of Mask Proposal Generation**
>  Table 3 shows the impact of mask proposal generation for the training-free setup. For example, with a CLIP-large backbone, we see a 43% drop in PQ (28.4 -> 16.1) when we use Grounding DINO + SAM instead of ground truth masks. This shows that the quality of mask proposal generation can have a significant impact on panoptic segmentation performance.
>
> References:
>
> 1. Johnson, Jeff, Matthijs Douze, and Hervé Jégou. "Billion-scale similarity search with GPUs." IEEE Transactions on Big Data 7.3 (2019): 535-547.
> 2. VS, Vibashan, et al. "Possam: Panoptic open-vocabulary segment anything." arXiv preprint arXiv:2403.09620 (2024).

---

### Official Review · Reviewer_5aAG · 2024-11-09

**Soundness:** 2
**Presentation:** 2
**Contribution:** 2
**Rating:** 5
**Confidence:** 3

**Summary:**

This paper presents an approach for open vocabulary panoptic segmentation by combining retrieval-based classification with standard image segmentation. In particular, the authors introduce a retrieval-augmented segmentation method that utilizes a database of paired image-text features. During inference to address the challenge of domain shift between masked and natural images, the model retrieves relevant features from this database using masked segment features from the input image as queries. This retrieval-based score is combined with scores from a vision-language model (CLIP) to enhance classification accuracy for unseen classes.

**Strengths:**

- The reviewer likes the integration of retrieval-based classification with CLIP-based scores to address the domain shift issues between masked images and natural images. It clearly improves the model's ability to recognize unseen classes without additional training.

- The paper's approach to construct a feature database from widely available paired image-text data is interesting. This setup enables adaptability without requiring pixel-level annotations.

- The paper is well-organized and well-written.

**Weaknesses:**

- The reviewer feels that the retrieval-based classification relies heavily on the quality and diversity of the feature database constructed from paired image-text data. If the database lacks sufficient variety or coverage, the method may struggle to classify certain unseen classes accurately, particularly in real-world scenarios with a wide range of objects.

- Further, the reviewer observed that the method uses Grounding DINO and SAM for generating masks in the training-free setup. However, SAM can produce suboptimal masks without human input which may degrade segmentation accuracy. This dependence on mask quality can limit the method’s effectiveness in fully automated settings.

- The authors may want to include methods such as ODISE for a more comprehensive analysis.

**Questions:**

N/A

---

> ### Author Response · Authors · 2024-11-23
> **Rebuttal to reviewer 5aAG**
>
> **Reliance on Feature Database**
> The reviewer expressed concern that the performance of the system depends on the quality and diversity of the feature database. We argue that the feature database is constructed from an aligned image-text dataset with class-level annotations, which is widely available on a very large scale. The YFCC100M dataset alone contains 100 million paired image-text samples. On the other hand, typical panoptic segmentation datasets require supervised annotations at a pixel level. Due to the difficulty of constructing a fully supervised panoptic segmentation dataset, the typical size and diversity of such datasets are limited (e.g., ADE20k with 20,000 samples and only 150 classes). Since our method can exploit image-text datasets with class-level annotations, we can easily improve the diversity and quality of the feature database by exploiting large-scale datasets such as YFCC100M. Using widely available class-level annotations to improve a system that typically requires pixel-level annotations is an important contribution of our proposed method.

---

> > ### Comment · Reviewer_5aAG · 2024-12-01
> > **Response to authors**
> >
> > Hi Authors,
> >
> > Do you have responses for my other two comments? I am willing to increase my score if proper responses are provided.

---

> > > ### Author Response · Authors · 2024-12-02
> > > **Rebuttal to reviewer 5aAG - 2**
> > >
> > > **Mask Quality in Training-Free Setup**
> > > SAM requires a point prompt or a bounding box input for high-quality mask generation. Since Grounding DINO is a strong open-set object detection model, it can generate the bounding box prompts required by SAM for thousands of novel classes. We refer the reviewer to the Grounding DINO paper for more details. While the quality of mask proposal generation in the training-free setup remains lower compared to the cross-dataset setup, we are not aware of any work that achieves higher performance in a fully training-free setup (without using pixel-level annotations in the same or cross-dataset settings).
> > >
> > > 1. Liu, Shilong, et al. "Grounding DINO: Marrying DINO with Grounded Pre-training for Open-set Object Detection." European Conference on Computer Vision. Springer, Cham, 2025.
> > >
> > > **ODISE**
> > >  We are happy to add ODISE as a baseline for comparison in the revised version of the paper. Incorporating a retrieval-based method with ODISE is beyond the scope of this work.
> > >
> > > We thank the reviewer for the thoughtful feedback, as it is very helpful for improving our work.

---

> ### Comment · Reviewer_5aAG · 2024-12-02
>
> Thank you for responding. Can you explain why comparison with ODISE is beyond the scope of this work? Aren't both of these methods aiming for open-vocab panoptic segmentation?

---

> > ### Author Response · Authors · 2024-12-02
> > **Rebuttal to reviewer 5aAG - 3**
> >
> > We did not mention comparison with ODISE is beyond the scope of work. Rather we mentioned that we are happy to add ODISE as a baseline for comparison. Quoting from the rebuttal above, `We are happy to add ODISE as a baseline for comparison in the revised version of the paper.` Our current implementation uses FC-CLIP as a backbone, adding a diffusion-based backbone with retrieval-based panoptic segmentation is beyond the scope of work. Relevant sentence from the above rebuttal `Incorporating a retrieval-based method with ODISE is beyond the scope of this work.` Apologies for any misunderstanding.

---

### Meta-Review · Area_Chair_9AzC · 2024-12-22

**Metareview:**

This paper presents a method for open vocabulary panoptic segmentation that merges retrieval-based categorization with conventional image segmentation. The manuscript was reviewed by four experts in the field. The recommendations are (2 x "3: reject, not good enough", 2 x "5: marginally below the acceptance threshold"). The reviewers raised many concerns regarding the paper, e.g., limited technical novelty, unclear motivation, unconvincing experimental evaluations, inadequate literature reviews, etc. Considering the reviewers' concerns, we regret that the paper cannot be recommended for acceptance at this time. The authors are encouraged to consider the reviewers' comments when revising the paper for submission elsewhere.

**Additional Comments On Reviewer Discussion:**

Reviewers mainly hold concerns regarding limited technical novelty (Reviewers GoTi), unclear motivation and statement (Reviewers 5aAG, X1Um, xAjY, GoTi), unconvincing experimental evaluations (Reviewers X1Um, xAjY, GoTi), inadequate literature reviews (Reviewer 5aAG, xAjY, GoTi). The authors' rebuttal could not fully address the above-listed concerns.

---

### Decision · Program_Chairs · 2025-01-22

Reject